# Extracellular Vesicle-Mediated Purinergic Signaling Contributes to Host Microenvironment Plasticity and Metastasis in Triple Negative Breast Cancer

**DOI:** 10.3390/ijms22020597

**Published:** 2021-01-09

**Authors:** Suzann Duan, Senny Nordmeier, Aidan E. Byrnes, Iain L. O. Buxton

**Affiliations:** 1Department of Pharmacology, School of Medicine, University of Nevada Reno, Reno, NV 89557, USA; senny.wong27@gmail.com (S.N.); aidanebyrnes@gmail.com (A.E.B.); 2Department of Medicine, College of Medicine, University of Arizona, Tucson, AZ 85724, USA; sduan@deptofmed.arizona.edu

**Keywords:** extracellular vesicles, exosomes, purinergic signaling, nucleoside diphosphate kinase, angiogenesis, metastasis, triple negative breast cancer

## Abstract

Metastasis accounts for over 90% of cancer-related deaths, yet the mechanisms guiding this process remain unclear. Secreted nucleoside diphosphate kinase A and B (NDPK) support breast cancer metastasis. Proteomic evidence confirms their presence in breast cancer-derived extracellular vesicles (EVs). We investigated the role of EV-associated NDPK in modulating the host microenvironment in favor of pre-metastatic niche formation. We measured NDPK expression and activity in EVs isolated from triple-negative breast cancer (MDA-MB-231) and non-tumorigenic mammary epithelial (HME1) cells using flow cytometry, western blot, and ATP assay. We evaluated the effects of EV-associated NDPK on endothelial cell migration, vascular remodeling, and metastasis. We further assessed MDA-MB-231 EV-induced proteomic changes in support of pre-metastatic lung niche formation. NDPK-B expression and phosphotransferase activity were enriched in MDA-MB-231 EVs that promote vascular endothelial cell migration and disrupt monolayer integrity. MDA-MB-231 EV-treated mice demonstrate pulmonary vascular leakage and enhanced experimental lung metastasis, whereas treatment with an NDPK inhibitor or a P2Y1 purinoreceptor antagonist blunts these effects. We identified perturbations to the purinergic signaling pathway in experimental lungs, lending evidence to support a role for EV-associated NDPK-B in lung pre-metastatic niche formation and metastatic outgrowth. These studies prompt further evaluation of NDPK-mediated EV signaling using targeted genetic silencing approaches.

## 1. Introduction

Breast cancer is the second leading cause of cancer death in women, with survival rates heavily influenced by disease stage, tumor grade, and receptor status. While the 5-year relative survival rate for women diagnosed with ductal carcinoma in situ remains high, women diagnosed with distant-stage breast cancer see a dramatic decrease in their odds of survival, dropping from a near 99% 5-year survival rate for localized breast cancer to only 27% [1]. This dichotomy in outcomes beckons the need for an improved understanding of the molecular mechanisms that guide breast cancer metastasis and support distant recurrence.

Functional interactions between primary tumor cells and the systemic host microenvironment orchestrate the formation of a metastatic milieu attuned to the establishment and expansion of circulating tumor cells (CTCs). Collectively termed the pre-metastatic niche (PMN), these tissue-specific sites are rich in tumor-supportive factors and demonstrate considerable plasticity and heterogeneity [2]. Cooperative and antagonistic signaling by tumor-secreted factors dictate PMN formation and abet CTCs in immune evasion, extravasation, and distant neocolonization. Such fine-tuned interactions with the distant microenvironment are observed during the early stages of primary tumor development and mediate the escape from indolence to metastatic outgrowth [3,4,5]. In recent years, tumor-derived extracellular vesicles (EVs) have attracted widespread attention for their roles in oncogenesis and are increasingly implicated in multiple cancer hallmarks [6,7]. Furthermore, tumor-derived EVs have been shown to facilitate paracrine-like and organotropic dialogue between primary tumor cells and the PMN [8,9].

Nucleoside diphosphate kinases (NDPKs) represent a ubiquitous and highly conserved class of multifunctional proteins transcribed from the NME/nm23 gene family and are known to be derived from multiple human cancer cell lines [10,11,12,13,14,15,16]. Originally identified by Steeg and colleagues as non-metastatic gene 23, *nm23-H1* (also known as *NME1* and NDPK-A) and *nm23-H2* (also referred to as *NME2* and NDPK-B) were among the first genes to be associated with a distinct metastasis suppressor function [10,11]. To date, NDPK-A and NDPK-B have remained the targets of exhaustive research to elucidate their suggested roles as metastasis suppressors. However, they are also associated with multiple cellular functions and may act as nucleoside phosphotransferases, histidine and serine/threonine protein kinases, transcriptional regulators, and DNA nucleases [12,13].

Extracellular vesicles derived from multiple human cancer cell lines [14,15,16,17,18,19,20] carry nucleoside diphosphate kinase A and B (EV-associated NDPK-A/B or eNDPK), implicated in promoting angiogenesis and pro-metastatic events extracellularly. The family of NDPK isoforms catalyzes the transfer of a γ-phosphate from a nucleoside triphosphate to a diphospho-nucleoside, and extracellular NDPK localized to the cell surface has been shown to regulate ADP/ATP levels by catalyzing ATP formation from ADP [21]. ATP is a primary agonist to the P2Y and P2X class of purinoreceptors commonly upregulated on tumor cells and vascular endothelium. ATP acting on these purinoreceptors elicits proinflammatory and immunosuppressive responses in the tumor microenvironment and has been shown to promote tumor growth and invasion [22,23,24,25,26]. We have previously demonstrated a pro-angiogenic role for NDPK-mediated activation of P2Y1 receptors on vascular endothelium. Mechanistically, ATP-mediated P2Y1 receptor activation transactivates vascular endothelial growth factor receptor 2 (VEGFR-2) and stimulates mitogenic signaling pathways [27,28,29]. Additionally, activation of P2Y1/2 receptors on endothelium induces the release of potent vasodilators such as nitric oxide and asserts a mechanistic path for enhanced CTC dissemination [30]. Furthermore, inhibition of NDPK transphosphorylase activity and/or blockade of P2Y1 receptor activation attenuate tumor growth and metastasis to the lung in an orthotopic murine model of triple negative breast cancer [31].

Extracellular vesicles derived from triple negative breast cancer cells are enriched in NDPK expression, suggesting a functional role for the enzyme in tumor EV-mediated communication with the PMN [14,15,16]. In support of its purinergic function, elevated ATP in the hundred micromolar range has been reported in the tumor microenvironment [32]. Moreover, ATP-mediated activation of the P2Y2 receptor was shown to stimulate PMN formation by inducing lysyl oxidase (LOX) and collagen crosslinking [33].

We investigated the hypothesis that NDPK associated with breast cancer EVs propagates purinergic signaling in the distant microenvironment to support PMN formation and metastasis. Here, we show that EVs elaborated by triple negative breast cancer cells are enriched in NDPK-B expression and phosphotransferase activity. We elucidate a functional role for eNDPK-B in the PMN by demonstrating eNDPK-mediated activation of endothelial remodeling in support of angiogenesis and vascular permeabilization. We provide further evidence to support eNDPK-B in metastatic outgrowth and suggest a role for eNDPK-B in modulating the lung vascular microenvironment in favor of PMN formation.

## 2. Results

### 2.1. Extracellular Vesicles Secreted by Non-Tumorigenic Human Mammary Cells and Triple Negative Breast Cancer Cells Demonstrate Classical Morphological and Molecular Features

EVs were purified by ultracentrifugation from the conditioned media of triple negative human breast cancer cells (MDA-MB-231) and from a non-tumorigenic mammary epithelial cell line (HME1). This method allows collection of the smallest of the extracellular vesicles, referred to as exosomes. Here, we use the term EVs to accommodate the possibility that the fraction collected is heterogeneous. TEM imaging revealed extracellular vesicles that were predominantly 50 to 80 nm in diameter. EVs possessed a characteristic phospholipid bilayer membrane and were relatively uniform in size and morphology (Figure 1A). Compared to cell lysates, EVs were enriched in classical exosome and EV markers, including CD9, CD63, CD81, ALIX, flotillin-1, and MFGE8. In contrast, cell lysates were enriched in calnexin, a cellular protein known to be absent in pure EV preparations (Figure 1B). EVs from MDA-MB-231 cells exhibited lower expression of the EV marker tsg101 compared to cell lysates, while HME1-derived EVs expressed low levels of calnexin. For downstream visualization and tracking studies, MDA-MB-231 cells were transduced with lentivirus expressing GFP- or RFP-labeled CD63. Expression of GFP and RFP EVs was confirmed by confocal laser scanning microscopy (CLSM) imaging (Figure 1C). EVs were also purified using ExoQuick-TC and further characterized by TEM and flow cytometry. EVs that were isolated using ExoQuick-TC were approximately 30–100 nm in diameter, displayed a phospholipid bilayer, and demonstrated positive labeling for CD63, CD9, and CD81 (Appendix A).

### 2.2. MDA-MB-231 EVs Are Enriched in NDPK-B Expression Compared to Non-Tumorigenic Mammary EVs

To evaluate whether HME1 and MDA-MB-231 EVs carry NDPK-A and NDPK-B, EVs were adsorbed to latex beads and analyzed by flow cytometry. HME1 and MDA-MB-231 EVs demonstrated positive labeling for CD63. Consistent with the previous western blot analysis, HME1-derived EV preparations exhibited higher CD63 expression compared to EV preparations from MDA-MB-231 cells (Figure 2A). Both populations demonstrated robust labeling for NDPK-B, whereas labeling for NDPK-A was undifferentiable from the isotype control (Figure 2B). Comparatively low surface expression of CD63 and NDPK-A/B on non-adsorbed EVs was further confirmed by positive antibody labeling (Figure 2C). As CD63 is the most frequently used marker to define bona fide EV populations, we used the expression of the tetraspanin as an internal quantitative control for total EV protein. Relative to CD63, NDPK-B expression was higher in MDA-MB-231 EVs compared to HME1 EVs (Figure 2D). Consistent with these observations, western blot analysis confirmed more robust CD9 expression in HME1 EVs compared to MDA-MB-231 EVs. Relative to CD9, NDPK-A and NDPK-B expression was approximately three-fold higher in MDA-MB-231 EVs. In line with the previous flow cytometry analysis, EVs derived from both cell lines possessed nearly 30-fold higher expression of NDPK-B compared to NDPK-A (Figure 2E).

### 2.3. MDA-MB-231 EVs Are Enriched in NDPK Phosphotransferase Activity

To determine whether elevated NDPK-B expression in EVs was correlated with higher NDPK phosphotransferase function, we performed a luciferin-luciferase ATP activity assay to measure NDPK-B phosphotransferase activity. Phosphotransferase activity was significantly higher in lysates compared to whole EVs, supporting the previous observation that NDPK is primarily associated with the intraluminal compartment. Additionally, MDA-MB-231 EV lysates demonstrated approximately three-fold higher transphosphorylase activity compared to HME1 lysates when assayed at equivalent concentrations (Figure 3A). We next tested whether ellagic acid (EA), a known inhibitor of NDPK transphosphorylase activity, could decrease the previously observed activity. Addition of EA reduced transphosphorylase activity of EVs in a dose-dependent manner, with a 10 μM concentration reducing activity by two- to three-fold (Figure 3B). The P2Y1 receptor antagonist MRS2179 was not examined in this experiment as the products of NDPK activity are expected to act on the P2 receptor, not on the enzyme per se. We further confirmed these properties in ExoQuick-TC purified EVs (Appendix A).

To test the robustness of these results, we evaluated NDPK phosphotransferase activity in EVs from two additional non-transformed human breast epithelial cell lines (MCF-10A and MCF-12F) and from three additional breast cancer cell lines (MDA-MB-157, MDA-MB-468, and MCF-7). NDPK phosphotransferase activity was highest in MDA-MB-231 and MCF-7 EVs; however, no appreciable difference was observed between MDA-MB-468 and MDA-MB-157 EVs compared to those of non-transformed cells. ATP generation was predominantly attributed to NDPK activity, as both basal ATP production and adenylate kinase activity were 10 times lower than NDPK activity in all EV populations (Figure 3C). In comparison to the previous EV lysates, whole non-lysed EVs exhibited minimal transphosphorylase activity, suggesting that NDPK activity is largely associated with the internal EV compartment (Figure 3D).

### 2.4. NDPK Inhibition and P2Y1 Receptor Antagonism Reverse the Pro-Migratory Effect of MDA-MB-231 EVs on Vascular Endothelial Cells

To investigate the functional role of MDA-MB-231 EVs on vascular endothelium, we assayed migration as an indicator of endothelial cell activation, remodeling, and angiogenesis. We evaluated the effect of three concentrations of EVs on the migration of human umbilical vein endothelial cells (HUVECs). At two of the three concentrations tested, EVs derived from MDA-MB-231 cells demonstrated greater migratory-initiating capability compared to those of HME1 cells (Figure 4A). Treatment of HUVECs with MDA-MB-231 EVs enhanced migration after 24 h compared to non-treated cells in low serum conditions. Inhibition of EV-associated NDPK transphosphorylase activity with EA significantly attenuated cell migration. As NDPK is known to activate downstream P2Y1 receptor signaling through the generation of extracellular ADP/ATP, we measured the effect of EV-mediated cell migration following a blockade of the P2Y1 receptor. Delivery of the P2Y1 receptor antagonists MRS2179, MRS2279, and MRS2500 abrogated EV-mediated endothelial cell migration (Figure 4B). The robustness of these events was further tested in primary VCAM-1^+^ murine lung endothelial cells (MLECs). Consistent with the previous observations, MDA-MB-231 EVs stimulated the migration of MLECs, and combination treatment with EA and MRS2179 reversed these events (Figure 4C).

### 2.5. NDPK Inhibition Rescues MDA-MB-231 EV-Mediated Permeabilization of Vascular Endothelial Cell Monolayers

As purinergic signaling has been implicated in vascular permeability leading to CTC dissemination, we next assayed the effect of varying MDA-MB-231 EV concentrations on endothelial cell monolayer integrity using a modified Boyden chamber approach (Appendix A). Treatment of human lung microvascular endothelial cells (HLMVECs) with MDA-MB-231 EVs increased monolayer permeabilization to FITC-labeled dextran compared to treatment with HME1 EVs and complete growth medium alone. Treatment with EA significantly reduced MDA-MB-231 EV-mediated permeabilization, whereas treatment with MRS2179 elicited a more moderate effect (Figure 5A). Similar results were observed in HUVECs following treatment with MDA-MB-231 EVs and EA. However, administration of MRS2179 alone or in combination with EA demonstrated no appreciable effect on reversing MDA-MB-231 EV-mediated HUVEC permeabilization (Figure 5B). These observations were consistent with a marked loss in monolayer integrity and the junction-associated protein β-catenin. In addition, EA was shown to localize to the cell membrane, suggesting an intracellular membrane-associated mechanism for junction regulation (Figure 5C).

### 2.6. MDA-MB-231 EVs Enhance Vascular Leakage in the Lung and NDPK Inhibition or P2Y1 Receptor Antagonism Ameliorates This Effect

We further evaluated the ability of MDA-MB-231 EVs to disrupt in vivo vascular integrity. Treatment duration was determined by following previous studies that identified PMN formation after three weeks of EV treatment [8,9]. EV exposure was subsequently prolonged to eight weeks to better mimic physiological metastasis. Following three- or eight-week treatment with MDA-MB-231 EVs, mice were injected systemically with Evans Blue dye (EBD) and lungs were imaged to visualize extravasated dye. Three-week MDA-MB-231 EV infused mice demonstrated enhanced pulmonary vascular leakage, as indicated by the presence of EBD surrounding blood vessels. Subsequent quantitation of EBD signal intensity showed a trending but nonsignificant increase in the total EBD signal in MDA-MB-231 EV-treated lungs compared to the control group (Figure 6A). Similarly, the perfused lungs of eight-week MDA-MB-231 EV-treated mice demonstrated higher levels of EBD signal compared to PBS treatment. Treatment with EA or MRS2179 individually reduced EBD signal in lung tissues; however, combination treatment showed no appreciable effect on EBD infiltration (Figure 6B). We further measured the absorbance of extracted EBD from perfused lungs. Consistent with the previous observations, EA or MRS2179 treatment reversed MDA-MB-231 EV-mediated vascular permeabilization, as indicated by a decreased amount of extravasated dye in lung tissue (Figure 6C).

NDPK-mediated activation of P2Y1/2 receptors on endothelium is known to stimulate pro-inflammatory and vaso-dilative factors, including nitric oxide synthase (NOS) and cyclooxygenase (COX). We mimicked the conditions used in previous in vitro assays and measured NOS and COX expression following delivery of EVs with a high dosage of inhibitors for three hours, or with a lower drug dosage for 24 h. Comparison between these conditions provides a broader understanding of the immediate and long-term signaling effects mediated by EVs. The 3- and 24-h treatment of HUVECs with MDA-MB-231 EVs led to a moderate, non-significant increase in eNOS expression, whereas treatment with a high dosage of MRS2179 (100 μM) attenuated this effect. Additionally, the 24-h treatment with a lower dosage of EA (10 μM) or combination drug treatment significantly decreased COX-1, COX-2, and iNOS expression, whereas three-hour treatment with a higher dosage of both drugs (100 μM) induced a three-fold increase in COX-2 expression (Appendix A).

### 2.7. Attenuation of eNDPK Activity or P2Y1 Receptor Activation Blunts the Development of Lung Metastases in an Experimental Metastasis Model

We next considered whether enhanced vascular permeability promotes metastasis of CTCs to the lung. Following eight-week treatment, mice were injected systemically with MDA-MB-231-Luc+ cells expressing CD63-GFP and metastasis to the lungs, liver, and brain was evaluated 30 days later (Figure 6D). While bioluminescence was detected in all mouse lungs, no signal was detected in the liver or brain, indicating the absence of overt metastases in these tissues (Appendix A). Lungs from mice treated with MDA-MB-231 EVs displayed higher bioluminescence intensity and luciferase protein expression compared to those of the control group, suggesting a greater presence of colonized MDA-MB-231 cells. Treatment with EA or MRS2179 reduced lung bioluminescence intensity and luciferase expression; however, collective administration of these drugs appeared to elicit no beneficial effect (Figure 6E,F). These observations were concomitant with a slight increase in NDPK-A/B levels in three-week treated experimental mice; however, no difference was observed following prolonged MDA-MB-231 EV treatment (Appendix A).

Intriguingly, prolonged MDA-MB-231 EV exposure leading to vascular remodeling and metastatic outgrowth was not associated with a significant change in circulating angiogenic cytokines that have been previously suggested in PMN formation, including VEGF-A. However, MDA-MB-231 EVs induced significant fluctuations in host production of immuno-modulatory cytokines, including suppression of TNF-α, IL-6, IL-10, and induction of the chemokines CCL2, CCL3, and CCL5 (Appendix A).

### 2.8. Proteomic Analysis of the Lung Following MDA-MB-231 EV Treatment Identifies Purinergic Events Known to Support Pre-Metastatic Niche Formation

To identify potential changes in the lung proteome that led to the previous functional observations, we performed mass spectrometry on three-week treated lungs bearing tandem mass tags. We identified 36 proteins that were differentially expressed between control and experimental cohorts at the *p* < 0.05 significance level with log_2_ fold-change > 0.5 (Figure 7A). A Principal Component Analysis of statistically significant proteins with raw *p*-values of *p* < 0.01 shows a distinct separation between control and experimental groups, with the first component capturing 65% of the experimental variance (Figure 7B).

Pathway ontology analysis on differentially expressed proteins (*p* < 0.05) with log_2_ fold-change over 0.5 was performed using Ingenuity Pathway Analysis (IPA), iPathwayGuide (IPG), and STRING software. Proteins with the greatest log_2_ fold-change were identified and ranked (Appendix A). All three software tools identified perturbations to the coagulation system and immune response, whereas iPG and STRING identified the involvement of hematopoietic cell lineage, ECM-receptor interaction, and platelet activation and aggregation pathways. Pathways and processes unique to iPG include wound healing, cell adhesion, and fibrinogen binding (Figure 7C). Additionally, activated cellular components include the integral, internal, and external components of the plasma membrane, indicating internalization and/or membrane receptor interaction with EVs. Relevant canonical pathways identified by IPA include caveolar-mediated endocytosis, coagulation, hypoxia inducible factor (HIF-1α) signaling, and P2Y purinergic receptor signaling (Figure 7D). The validity of this experiment was confirmed by western blot of selected differentially expressed proteins identified by mass spectrometry (Appendix A). Collectively, these analyses identify purinergic events known to support PMN formation.

To add further depth to these experiments, we assessed proteomic cargo associated with MDA-MB-231 EVs and EVs from the non-tumorigenic MCF-12F cell line. Significant differentially expressed proteins with over three-fold enrichment were identified in MCF-12F and MDA-MB-231 vesicles (Appendix A). Processes associated with MDA-MB-231 EVs include cell migration, ECM disassembly, and positive regulation of angiogenesis. In support of the previous observations, these findings suggest that EV cargoes reflect parent cell lineage and propagate their malignant features.

## 3. Discussion

Since its identification 30 years ago as the first novel metastasis suppressor gene [10], *nm23* and its prognostic value in cancer remain controversial. Reduced *nm23-H1* mRNA and NDPK expression levels have long been reported in solid tumors, and loss of *nm23* correlates with disease progression and metastasis in several cancers. Investigations by our lab and others into extracellular *nm23* illuminate a diverging plotline for its role in oncogenesis. Among these, elevated *nm23* expression was reported in sera of patients with breast cancer [31], colorectal cancer [34,35], neuroblastoma [36], renal carcinoma [37], and hematological malignancies [38]. Metastatic breast cancer cells of human and murine origin secrete elevated levels of NDPK-A and -B isoforms compared to non-tumorigenic breast cells, suggesting a functional role for secreted NDPK in metastasis [39,40]. However, the mechanism by which NDPK-A/B is elaborated into the extracellular milieu remains unclear. Here, we showed that the highly metastatic and triple negative MDA-MB-231 cell line propagates functional NDPK-B phosphotransferase activity through the secretion of EVs.

We corroborate findings by several other groups showing pro-angiogenic and endothelium-remodeling properties associated with tumor-secreted EVs [41,42]. In accordance with the Nucleotide Axis Hypothesis first proposed by Buxton and colleagues [21], we present a role for NDPK associated with EVs in mediating P2Y1 receptor-dependent endothelial cell activation and remodeling. Interestingly, our observations suggest that dual inhibition of NDPK and P2Y1 receptor activation diminishes the beneficial effects observed when drug compounds are delivered individually. Further work is required to elucidate whether these events are attributed to the observed induction of COX-2 following high dosage treatment in endothelial cells, or whether P2Y2R and other purinoreceptors exert a compensatory effect. Indeed, activation of P2Y2 receptor by eATP has been shown to promote PMN formation [33] and enhance the invasiveness of breast cancer cells via crosstalk with endothelium [24]. Thus, blocking dual P2Y1/2 receptor activation may potentiate their individual anti-metastatic effects.

Cancer-derived extracellular vesicles have been shown to be enriched in a variety of both pro-angiogenic and angio-static factors. While we did not presently explore the contributions of the latter, it is well established in the literature that such antagonistic signaling factors may influence endothelial cell activation and suppression. Based on our observations, the pro-migratory effect induced by EV NDPK may be opposed by other anti-angiogenic factors at higher EV concentrations. For instance, our identification of thrombospondin-1 as a significantly enriched factor in breast cancer EVs lends evidence to support such antagonistic signaling.

Importantly, the pharmacological studies presented here do not address how fluctuations in NDPK protein levels influence downstream EV-mediated signaling. Delivery of siRNA or shRNA to deplete NDPK in breast cancer-secreted EVs provides a targeted silencing approach to determine whether the pro-angiogenic activity of EVs is indeed largely attributed to the NDPK protein. Such studies are supported by our observations that pharmacological inhibition of NDPK phosphotransferase activity and blockade of the downstream P2Y1 purinoreceptor independently blunt pro-angiogenic signaling by MDA-MB-231 EVs. In line with these observations, total phosphotransferase activity in breast cancer-secreted EVs is generated by NDPK, and not by adenylate kinase or other ATP-generating enzymes. Future application of genetic silencing approaches would enable a comprehensive dissection of these signaling mechanisms and lend further rigor to these observations.

As NDPK phosphotransferase activity was predominantly associated with the intraluminal EV compartment, there remains the question of how NDPK mediates extracellular nucleotide homeostasis. One possibility exists whereby NDPK associated with EVs is released into the cytosol of the recipient cell following direct membrane fusion. Previous reports identify a function for membrane-associated NDPK as a nucleotide reservoir and as a required component for caveolin and G protein-mediated VEGFR-2 activation [43,44]. Direct fusion of EVs may also facilitate ectopic NDPK expression on the recipient cell surface (Figure 8). Such a mechanism is supported by previous reports of NDPK localizing to caveolae and binding to the membrane [45,46]. As these membranous components are known to be recycled during exosome and EV biogenesis, it, thus, remains a possibility that NDPK associates with the EV membrane and is transferred to recipient cell membranes. Once ectopically expressed, NDPK can propagate its phosphotransferase activity to activate adjacent P2Y1/2 receptors co-expressed on endothelium. This proposed mechanism is congruent with previous observations by our group that NDPK is ectopically expressed on vascular endothelium.

We identified MDA-MB-231 EV-mediated perturbations to the P2Y and HIF-1α signaling pathways, lending evidence to support purinergic regulation of HIF-1α–LOX signaling in PMN formation [33]. Of further interest were pathways involved in platelet activation, aggregation, and wound healing, as these responses are mediated by purinergic signaling and are implicated in PMN formation and metastasis [47,48,49,50]. These analyses suggest that MDA-MB-231 EVs activate platelets, potentially through eNDPK-mediated activation of P2Y1 and P2Y12 receptors expressed on platelets. Further investigation into the role of eNDPK in modulating these purinergic pathways may reveal additional downstream effectors that can be targeted therapeutically to blunt metastatic progression.

Finally, our observations raise a previously unvisited question concerning the homeostatic dynamic of EV-mediated NDPK signaling. In addition to NDPK, several other ectoenzymes mediate purine metabolism and homeostasis in the tumor microenvironment, including CD39, CD73, adenosine deaminase, and adenylate kinase. The importance of their collective roles in the tumor microenvironment is undermined by the fact that both adenosine and ATP are present at remarkably high levels in tumor interstitium compared to in non-tumor bearing tissues [32,51]. Moreover, adenosine and ATP mediated activation of purinergic receptors has been reported to play a role in modulating immune cell function, angiogenesis, and the inflammatory response [52]. While the studies presented here sought to investigate EV-mediated NDPK signaling in isolation, further work examining the relationship between NDPK and other ectonucleotidases will shed additional light on homeostatic signaling within the tumor microenvironment.

## 4. Materials and Methods

### 4.1. Drug Reagents

MRS compounds and ellagic acid were purchased from Tocris Bioscience (Bio-Techne, Avonmouth, Bristol, UK). For the animal studies, ellagic acid was purchased from MilliporeSigma (Burlington, MA, USA). MRS compounds and ellagic acid were used to antagonize the P2Y1 receptor and inhibit NDPK, respectively.

### 4.2. Cell Culture of Human Cell Lines

Primary human umbilical vein endothelial cell (HUVEC) and human lung microvascular endothelial cell (HLMVEC) lines (Lifeline Cell Technology, Frederick, MD, USA) were grown in VascuLife-VEGF and Vasculife-MV media, respectively (Lifeline Cell Technology). Media was supplemented with the respective growth factor kits and 10% EV-depleted FBS (Atlanta Biologicals, Flowery Branch, GA, USA). MDA-MB-231, MDA-MB-231-Luc+, MDA-MB-157, MDA-MB-468, MCF-7, MCF-10A, MCF-12F, telomerized human mammary epithelial cells (HME1/HMECs), and 3T3 fibroblast cells were purchased from ATCC (Manassas, VA, USA). MDA-MB-231 and MDA-MB-231-Luc+ cells were transduced with lentivirus vector expressing a GFP or RFP tag fused to the membrane-associated tetraspanin CD63 (System Biosciences, Palo Alto, CA, USA). Breast cancer cells were cultured in Dulbecco’s Modified Eagle Medium (DMEM) supplemented with 2% penicillin-streptomycin and 10% EV-depleted FBS. FBS was filtered through a 0.2 μm membrane and ultracentrifuged at 100,000× *g* for 24 h at 4 °C to deplete EVs. Non-transformed breast epithelial cell lines were cultured in Human Mammary Epithelial Cell Medium (HuMEC) supplemented with 2% penicillin-streptomycin, 10% EV-depleted FBS, and HuMEC supplement kit excluding bovine pituitary extract (Thermo Fisher Scientific, Waltham, MA, USA). Cells were cultured under 5% CO_2_ in room air at 37 °C.

### 4.3. Neonatal Pulmonary Endothelial Cell Isolation

Approval was granted by the Institutional Animal Care and Use Committee prior to animal use. Methods were adapted from Sobczak et al. [53]. Digested lung tissue was incubated with Dynabeads™ (Thermo Fisher Scientific) coated with intercellular adhesion molecule (ICAM-1, ICAM-2) and vascular cell adhesion molecule (VCAM-1) antibodies conjugated to Alexa Fluor 680 (BioLegend, San Diego, CA, USA). Sorted cells were cultured in Endothelial Growth Media (EGM-2) (Lonza, Basel, Switzerland) on 2% gelatin-coated chamber slides. Cells were fixed in 4% paraformaldehyde (PFA), incubated in anti-wheat germ agglutinin conjugated to Alex Fluor 488 (1:1000, Thermo Fisher Scientific), and imaged by confocal laser microscopy (CLSM) at 200× magnification.

### 4.4. Extracellular Vesicle Isolation

Cells were cultured for 4–5 days in DMEM with 10% EV-depleted FBS. Conditioned media was centrifuged at 3000× *g* for 10 min and the supernatant was filtered through a 0.2 μm membrane before centrifuging again at 20,000× *g* for 30 min. The supernatant was ultracentrifuged at 100,000× *g* for 70 min and the EV pellet was washed in PBS before repeating the spin. EVs were re-suspended in sterile PBS and total protein was quantified using an EZQ assay (Thermo Fisher Scientific). This method allows us to collect the smallest of the extracellular vesicles, referred to as exosomes. We have used the term EVs to accommodate the possibility that the fraction collected is heterogeneous. For data presented in Appendix A, EVs were isolated using ExoQuick-TC (System Biosciences, Palo Alto, CA USA) according to the manufacturers’ instructions. For experimental metastasis study only, both ultracentrifugation- and ExoQuick-TC-purified EVs were used in order to meet experimental demand (four weeks treatment with each).

### 4.5. Transmission Electron Microscopy (TEM)

Transmission electron microscopy grids were prepared according to Thery et al. [35]. Grids were analyzed by the University of Nevada, Reno Imaging Core using the Philips CM10 transmission electron microscope and a GATAN BioScan 792 digital scanner.

### 4.6. EV Analysis with Flow Cytometry

FACS protocol by Thery et al. [54] was used for flow cytometry analysis of EVs adsorbed to latex beads. Briefly, purified EVs were incubated with 3.9 μm latex beads (Thermo Fisher Scientific) and rocked overnight at 4 °C. Beads with adsorbed EV proteins were blocked in 100 mM glycine for 30 min at room temperature, then centrifuged for 3 min at 1500× g. The bead pellet was washed three times with 0.5% BSA in PBS before incubating in 1:200 rabbit anti-human NM23A (Abcam, Cambridge, MA, USA), NME2 (Abcam), or matched rabbit isotype primary antibodies (Abcam) for 60 min at 4 °C. Beads were washed twice and incubated in PE-conjugated donkey anti-rabbit secondary antibody (Biolegend) for 30 min at 4 °C. Samples were washed twice, resuspended in 0.5% BSA and analyzed on the BD LSR II Flow Cytometer (BD Biosciences, San Jose, CA, USA) at the University of Nevada, Reno Flow Cytometry Core. For surface labeling, EVs were incubated for 30 min at room temperature with PE-conjugated mouse IgG1 isotype (Biolegend), CD63 (Biolegend), or NDPK-A (US Biologicals, Salem, MA, USA) antibodies diluted in 0.5% BSA in PBS. Data were analyzed using FlowJo software version 10.5.3 (FlowJo LLC, Ashland, OR, USA). For NDPK-B quantitation, side scatter area (SSC-A, *y*-axis) was plotted against PE area (PE-A, *x*-axis) on a log scale. Gates were manually drawn to exclude 98–99% of sample isotype signal intensity and copied to respective plots of primary antibody labeling. Gates captured approximately 95% of total NDPK-B signal. Overlapping values from respective isotype controls were subtracted from gated NDPK-B and CD63 values and NDPK-B was normalized to CD63 intensity and plotted using Graphpad Prism version 8 (Graphpad, San Diego, CA, USA); *n* = 3 sample replicates.

### 4.7. Western Blot Analysis

For NDPK expression analysis, EV protein was mixed with SDS sample buffer and 1% triton-X before heating at 70 °C for 10 min. Samples were loaded onto a 4–20% pre-cast TGX polyacrylamide gel (Bio-Rad Laboratories, Hercules, CA, USA) and run at 200V for approximately 45 min. Proteins were transferred onto a 0.2 μm nitrocellulose membrane using the Turboblot transfer system. Membranes were blocked in 5% BSA in PBS with 0.05% Tween-20 and incubated overnight at 4 °C in NM23A (Abcam), NME2 (Abcam), and CD9 (Biolegend) primary antibodies diluted 1:250. Membranes were imaged on the LI-COR Odyssey with Image Studio software (LI-COR, Lincoln, NE, USA). For 10 μg of EV, protein and cell lysate were prepared as previously described. HUVEC and lung lysates were prepared as described previously, except in reducing conditions. Antibody dilutions and sources are listed in Appendix A.

### 4.8. Wes Protein Assay

HUVECs were treated for three hours with 100 ng/mL HME1 or MDA-MB-231 EVs, alone or with 100 uM MRS2179 and ellagic acid. Equal protein lysate concentrations (0.2 mg/mL) were run on ProteinSimple Wes™ according to manufacturer instructions (Biotechne, Minneapolis, MN, USA). eNOS (1:100, Santa Cruz Biotechnology, Dallas, TX, USA), iNOS (1:50, Santa Cruz Biotechnology), COX-2 (1:50, Cell Signaling Technologies, Danvers, MA, USA), and GAPDH (Cell Signaling Technologies) antibodies were used to measure protein expression.

### 4.9. NDPK Transphosphorylation Activity Assay

EV lysates were prepared in 10% triton X-100 in room Air Krebs (RAK) buffer without CaCl_2_. Recombinant NDPK-A, NDPK-B, and ATP were serially diluted into standards and incubated with 10 µM ADP and 30 µM UTP before quenching with 0.1 M HCl in prepared buffer. Standards were run with whole EVs or EV lysates. The pH was neutralized to 7.4 and equal volumes of luciferin-luciferase ATP detection buffer were added. Luminescence was measured using the Hidex Chameleon V plate reader with MikroWin 2000 software version 4.43 (Hidex, Turku, Finland).

### 4.10. Transwell Migration Assay

Transwell inserts with 3 μm pores and 12 mm diameter (Corning, Corning, NY, USA) were coated with rat-tail collagen (Corning). HUVECs were grown to 75% confluence and serum-starved for 24 h prior to seeding in top chambers containing VascuLife media with 2% EV-depleted FBS (2.5 × 10^5^ cells per well). Assay was initiated by adding EVs (100 μg/mL or 1 ug/mL) to bottom chambers and P2Y1 receptor antagonists (10 μM MRS2179, 100 nM MRS2279, 10 nM MRS2500) or an NDPK inhibitor (10 μM EA) to top chambers (*n* = 3). After 24 h, membranes were processed and imaged at 100× magnification using the Keyence BZ-X700 automated microscope. Migrated cells were quantified using the BZ-X700 Analyzer Software (Keyence, Osaka, Japan). Three images at the center of each well were quantified and averaged to generate a count for each well. The assay was repeated with MLECs in EGM-2 media (Lonza).

### 4.11. Transwell Permeability Assay

Methods were adapted from Martins-Green et al. [55]. Transwell inserts were coated with growth factor-reduced Matrigel^®^ (Corning) and seeded with HUVECs and HLMVECs in Vasculife-VEGF or Vasculife-MV media (1 × 10^5^ cells/well). A second monolayer was seeded 48 h later. The assay was initiated 48 h later with the addition of 10 μg/mL of the final concentration of FITC-dextran sulfate (MilliporeSigma) and 100 ng/mL of the EVs to the lower chambers, while inhibitors (100 μM EA, MRS2179, or both) were added to top chambers. Media was collected from the top chambers at indicated time points and fluorescence was measured with 485 nm excitation/535 nm emission using the Hidex Chameleon V plate reader and MikroWin 2000 software version 4.43 (Hidex). *n* = 6.

### 4.12. Imaging of Endothelial Barrier

HLMVECs were seeded and treated as described in the previous experiment, but without the addition of FITC-dextran. Cells were treated for three hours, washed with PBS, fixed in 4% PFA, permeabilized with 1% Triton-X, and incubated overnight in 1:250 FITC-phalloidin (Thermo Fisher Scientific) or β-catenin antibodies (Thermo Fisher Scientific). Whole membranes were mounted with Vectashield containing DAPI (Vector Laboratories, Burlingame, CA, USA). Slides were imaged by CLSM at 200× magnification.

### 4.13. Confocal Laser Scanning Microscopy (CLSM)

Image acquisition and analysis were performed using an Olympus IX81 Fluoview confocal microscope and FV10-ASW software version 4.02, Windows 7 (Olympus America, Inc., Melville, NY, USA). All image acquisition settings and post hoc image adjustments (i.e., brightness, contrast, and LUT) were applied globally for each experiment to ensure accuracy in comparison between controls and experimental groups.

### 4.14. Animal Studies

The University of Nevada, Reno Institutional Animal Care and Use Committee approved all studies and procedures prior to animal use. NCr/SCID mice were purchased from Charles River Laboratories (Wilmington, MA, USA) and bred on site. All subsequent litters were age matched for experiments.

### 4.15. TMT-Labeled Mass Spectrometry

Eight-week-old female SCID mice were injected by tail vein with 5–10 μg of purified MDA-MB-231 EVs three times per week for three weeks. Whole lungs were collected, powdered, and reconstituted in NP-40 lysis buffer. Samples were submitted to the Nevada Proteomics Center for mass spectrometry analysis. The methodology and analysis are described in detail in Appendix A.

### 4.16. Evans Blue Dye (EBD) Extravasation Study

Ten-week-old SCID mice were injected by tail vein with PBS vehicle or 5–10 μg of purified MDA-MB-231 EVs three times per week for eight weeks. MRS2179 was injected by tail vein with EVs (8.5 μg) while ellagic acid (EA) was administered via drinking water (120 μg/mL equivalent to 5 ± 0.7 mL/day per mouse [29]). A final group received both drugs (Combo). After eight weeks, mice were injected by tail vein with 2% EBD and re-caged for three hours (*n* = 4–6 per group). Lungs were collected following cardiac perfusion with PBS. Right lung lobes were incubated in *N,N*-dimethylformamide to extract EBD. Absorbance at 610 nm was measured using MikroWin 2000 software (Hidex). Left lung lobes were cryo-sectioned to 10 μm thickness and imaged by CLSM at 100× magnification.

### 4.17. Experimental Metastasis Study

Ten-week-old SCID mice were treated for eight weeks as previously described (*n* = 3–5 per group). Following treatment, mice were injected by tail vein with 1 × 10^5^ MDA-MB-231-luc^+^ cells engineered to express GFP-tagged CD63. Mice were imaged weekly using Living Image Software Version 4.5 and the Lumina III In Vivo Imaging System (IVIS, PerkinElmer, Waltham, MA, USA). Thirty days later, whole lungs, liver, and brain were removed, incubated in D-luciferin, and imaged.

### 4.18. Statistical Analysis

One- or two-way analysis of variance (ANOVA) was used for assays comparing three or more groups. Unpaired two-tailed Student’s *t*-test or Mann-Whitney nonparametric test were used for all other assays, comparing between two groups, as indicated in the figure legends. Tukey’s post-test was applied when appropriate, and significance was defined as follows: * = *p* < 0.05, ** = *p* < 0.01, *** = *p* < 0.001.

## Figures and Tables

**Figure 1 ijms-22-00597-f001:**
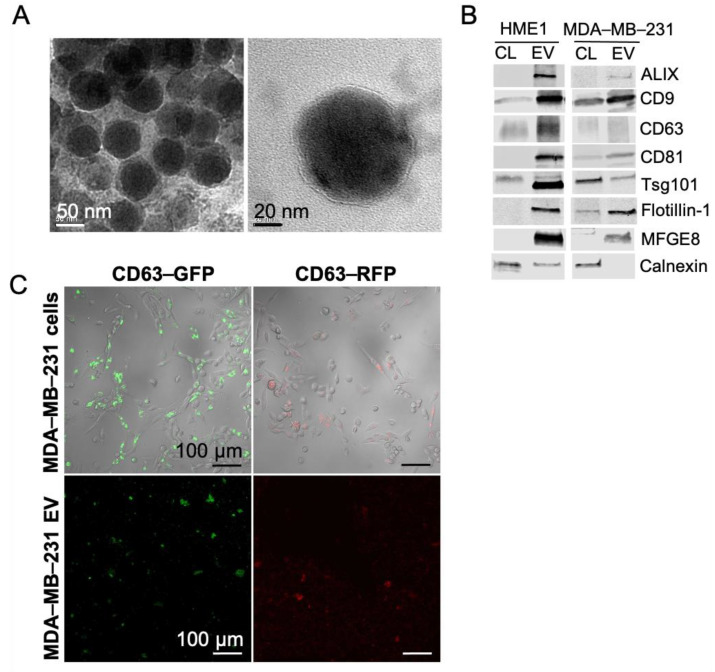
Characterization of HME1 and MDA-MB-231 EVs by transmission electron microscopy, confocal microscopy, and western blot. (**A**) TEM images of MDA-MB-231 EVs stained with uranyl-acetate. Scale: 50 and 20 nm. (**B**) Western blots of classical exosome and EV protein markers in HME1 and MDA-MB-231 EV lysates. Calnexin was evaluated as a cell-enriched marker. (**C**) CLSM images of MDA-MB-231 cells and respective purified EVs expressing GFP or RFP-labeled CD63. Scale: 100 μm.

**Figure 2 ijms-22-00597-f002:**
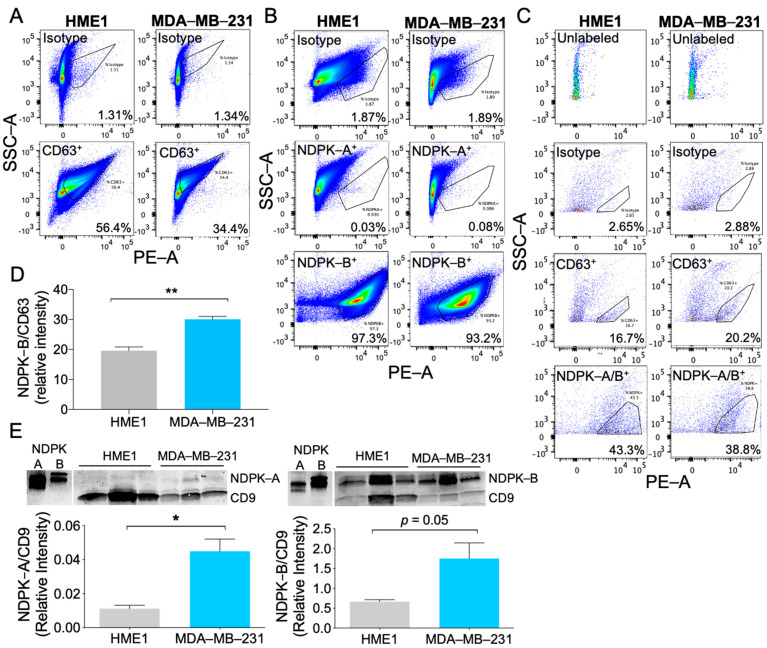
Non-tumorigenic breast epithelial cells and MDA-MB-231 breast cancer cells release EVs carrying NDPK-A and NDPK-B. (**A**) Flow cytometry analysis of HME1 and MDA-MB-231 EVs adsorbed onto latex beads and immuno-labeled for CD63 and (**B**) NDPK-A and NDPK-B. (**C**) Flow cytometry analysis of non-adsorbed HME1 and MDA-MB-231 EVs labeled for surface expression of CD63 and NDPK-A/B. Plots show the side scatter area (*y*-axis) versus the PE-fluorescence area (*x*-axis) on a log scale. (**D**) Quantitation of positive NDPK-B labeling in HME1 and MDA-MB-231 EVs normalized to CD63 intensity and reflecting background subtraction of respective isotype control. Mean ± S.E.M; ** *p <* 0.01 by two-tailed Student’s *t*-test, *n* = 3 sample replicates. (**E**) Western blot analysis of NDPK-A and NDPK-B expression normalized to CD9 in HME1 and MDA-MB-231 EV lysates. Lanes represent sample replicates consisting of EV lysates from independent isolations. Cross-reactivity of antibodies was confirmed using recombinant NDPK-A/B protein. Mean ± S.E.M; * *p <* 0.05 by two-tailed Student’s *t*-test, *n* = 3.

**Figure 3 ijms-22-00597-f003:**
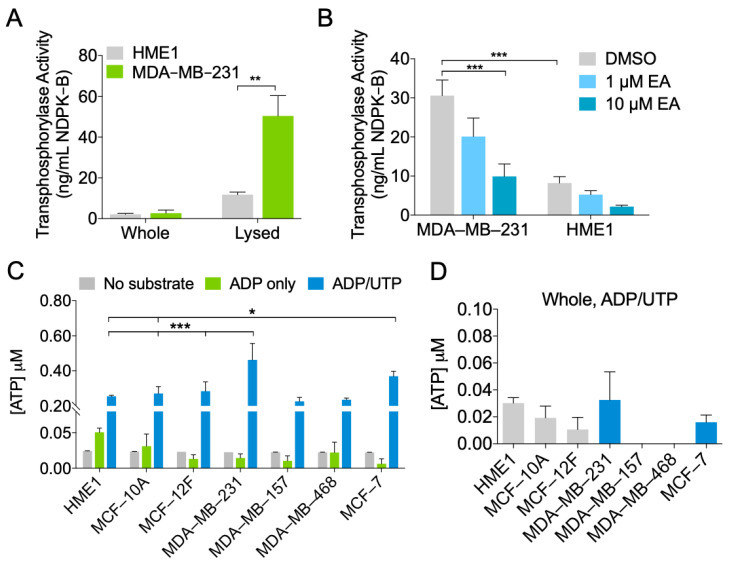
MDA-MB-231 EVs are enriched in NDPK transphosphorylase activity. (**A**) Measurement of NDPK transphosphorylase activity in whole and lysed EVs secreted by HME1 and MDA-MB-231 cells. *n* = 3 sample replicates. Mean ± S.E.M. ** *p* < 0.05 by two-tailed Student’s *t*-test. (**B**) Measurement of NDPK transphosphorylase activity in MDA-MB-231 and HME1 EVs treated with ellagic acid (EA) or with DMSO vehicle. *n* = 3 sample replicates. Mean ± S.E.M. *** *p* < 0.001 by two-way ANOVA. (**C**) EVs derived from multiple non-transformed and breast cancer cell lines were incubated with ADP/UTP substrate, ADP only, or no substrate, and ATP production was measured. (**D**) NDPK transphosphorylase activity of whole non-lysed EVs was measured. Mean ± S.E.M. * *p* < 0.05, *** *p* < 0.001 by two-way ANOVA.

**Figure 4 ijms-22-00597-f004:**
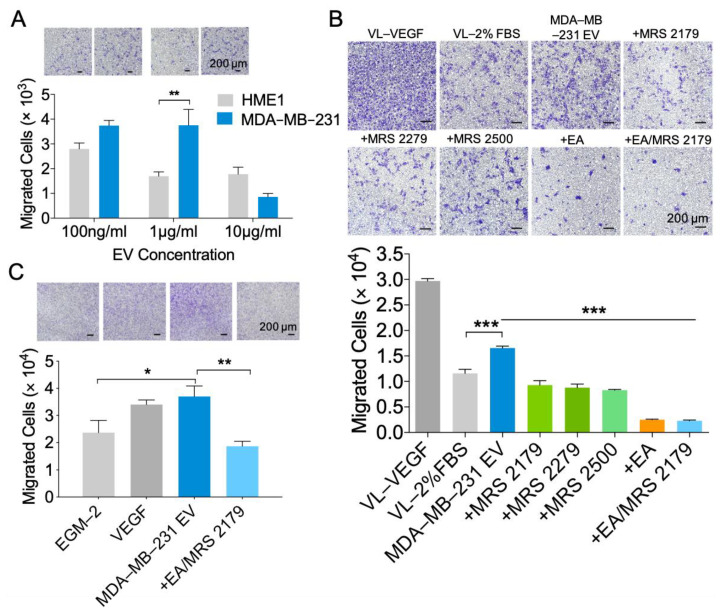
NDPK inhibition and P2Y1 receptor antagonism reverse MDA-MB-231 EV-mediated vascular endothelial cell migration. (**A**) Quantitation of HUVEC migration through collagen-coated transwells following a 24-h treatment with MDA-MB-231 EVs or HME1 EVs. Scale: 200 μm. *n* = 3. Mean ± S.E.M. ** *p* < 0.01 by nonparametric two-tailed Student’s *t*-test. (**B**) HUVEC migration through collagen-coated transwells following a 24-h treatment with complete growth medium (VL-VEGF), low serum (VL-2% FBS), MDA-MB-231 EVs (MDA-231 EV) with and without MRS2500, MRS2270, MRS2179, ellagic acid (EA), and ellagic acid with MRS2179. Scale: 200 μm. *n* = 3. Mean ± S.E.M. *** *p* < 0.001 by one-way ANOVA and Tukey’s post-test. (**C**) MLEC migration through collagen-coated transwells following a 24-h treatment with complete growth medium (EGM-2), VEGF, MDA-MB-231 EVs, and EVs with ellagic acid and MRS2179. Scale: 200 μm. *n* = 3. Mean ± S.E.M. * *p* < 0.05, ** *p* < 0.01 by one-way ANOVA and Tukey’s post-test.

**Figure 5 ijms-22-00597-f005:**
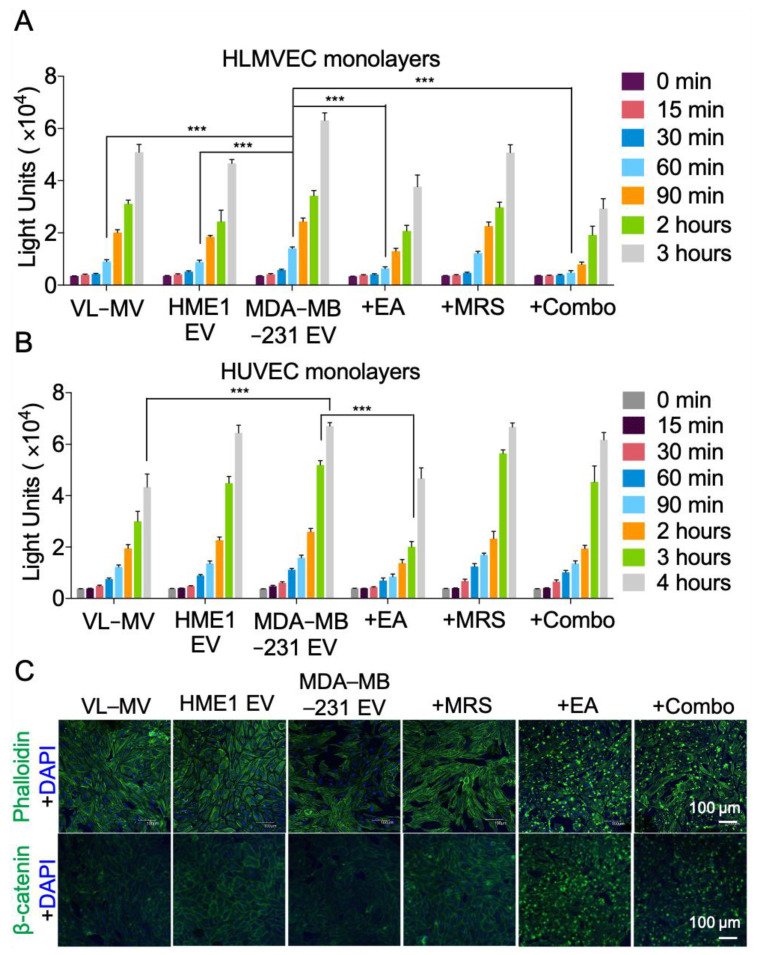
NDPK inhibition ameliorates endothelial monolayer permeabilization by MDA-MB-231 EVs. (**A**) Permeabilization of HLMVEC and (**B**) HUVEC monolayers to FITC-dextran was measured using Matrigel-coated transwell chambers. Cells were treated with complete growth medium (VL-MV or VL-VEGF) with and without HME1 EVs, MDA-MB-231 EVs, and MDA-MB-231 EVs with MRS2179, EA, or a combination of both drugs. Intensity of FITC-dextran in the top chambers was measured at indicated time points as an indicator of enhanced permeability. *n* = 6. Mean ± S.E.M. *** *p* < 0.001 by one-way ANOVA and Tukey’s post-test. (**C**) CLSM images of HLMVEC monolayers immuno-stained for actin and β-catenin expression. All CLSM laser acquisitions were kept consistent, except for the bottom two images on the right, where 488 nm laser was enhanced to better visualize cells in these two treatment conditions. Cells were counterstained with DAPI. Scale: 100 μm.

**Figure 6 ijms-22-00597-f006:**
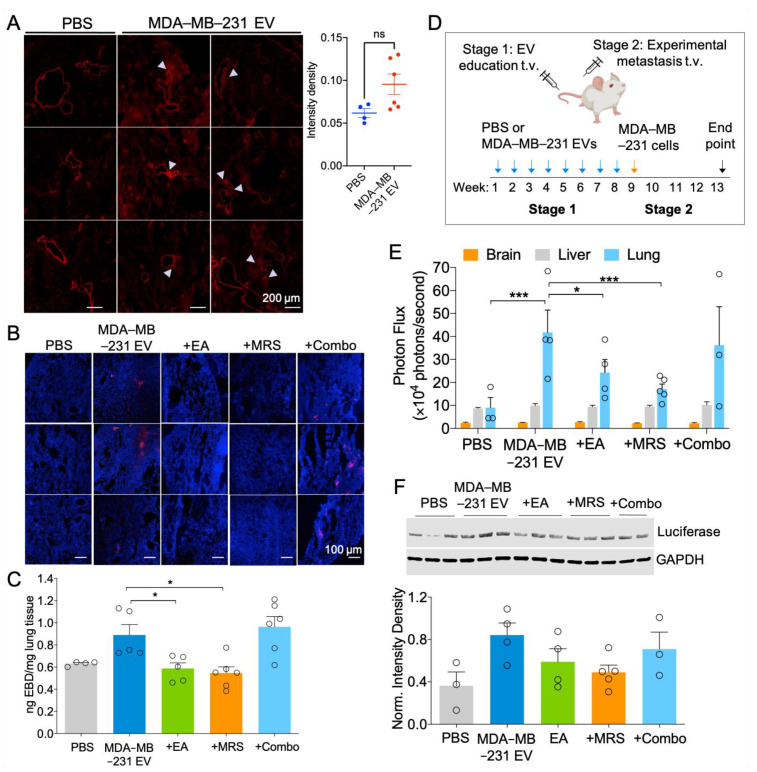
Inhibition of NDPK phosphotransferase activity or P2Y1 receptor blockade blunts MDA-MB-231 EV-mediated vascular leakage and experimental lung metastasis. (**A**) Representative images of lungs from mice treated for three weeks with MDA-MB-231 EVs or PBS vehicle and systemically injected with EBD emitting in the far-red spectrum. Arrows indicate regions of vascular leakage. *n* = 3 per group. Scale: 200 μm. Quantitation of total image intensity is also shown (circles represent *n* = 4–6 ROIs). (**B**) Representative CLSM images of lungs from mice treated for 8 weeks with PBS or MDA-MB-231 EVs in combination with drug inhibitors. Mice were systemically injected with EBD (red) and perfused to clear EBD from lung vasculature. Lung sections were counterstained with DAPI. Scale: 100 μm. (**C**) EBD was biochemically extracted from 8-week treated perfused mouse lungs and quantified by measuring absorbance at 610 nm. Circles represent *n* = 4–6 mice per group. Mean ± S.E.M. * *p* < 0.05 by one-way ANOVA and Tukey’s post-test. (**D**) Visual summary of experimental metastasis study design. (**E**) The brain, liver, and lungs of 8-week treated mice were imaged ex vivo for bioluminescence intensity to detect the presence of experimental MDA-MB-231-Luc^+^ metastases. Circles represent *n* = 3–5 mice per group. Mean ± S.E.M. * *p* < 0.05, *** *p* < 0.001 by two-way ANOVA. (**F**) Western blot analysis of luciferase expression relative to GAPDH in the right lung lobe of 8-week treated mice. Circles represent *n* = 3–5 mice per group. Data not significant by one-way ANOVA.

**Figure 7 ijms-22-00597-f007:**
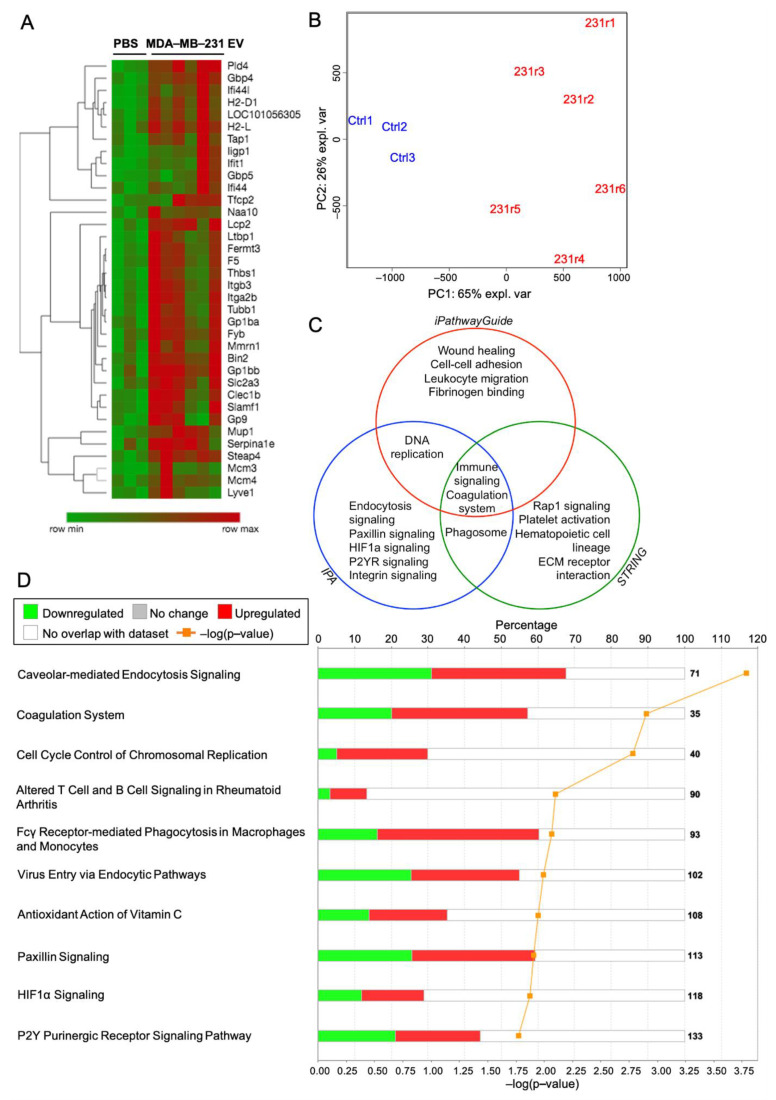
MDA-MB-231 EVs induce proteomic transformation in the lung and activate purinergic events associated with pre-metastatic niche formation. Mass spectrometry was performed on TMT-labeled lungs from mice treated for three weeks with PBS (*n* = 3) or MDA-MB-231 EVs (*n* = 6). (**A**) Heatmap with hierarchical clustering of differentially expressed proteins at the *p* < 0.05 significance level with log2 fold-change above 0.5. Row min and max reflect log2 fold-change values. (**B**) Principal component analysis of the top 107 differentially expressed proteins with raw *p*-values < 0.01. *n* = 3 for PBS-treated mice (Ctrl1-3) and *n* = 6 for MDA-MB-231 EV-treated mice (231r1-6). (**C**) Venn diagram of significant pathways, biological processes, and molecular functions identified by IPA, iPG, and STRING software. FDR correction was applied at the protein dataset level and during pathway analyses with significance defined at *p* < 0.05. (**D**) Top 10 canonical pathways perturbed by MDA-MB-231 EV treatment, as identified by IPA analysis.

**Figure 8 ijms-22-00597-f008:**
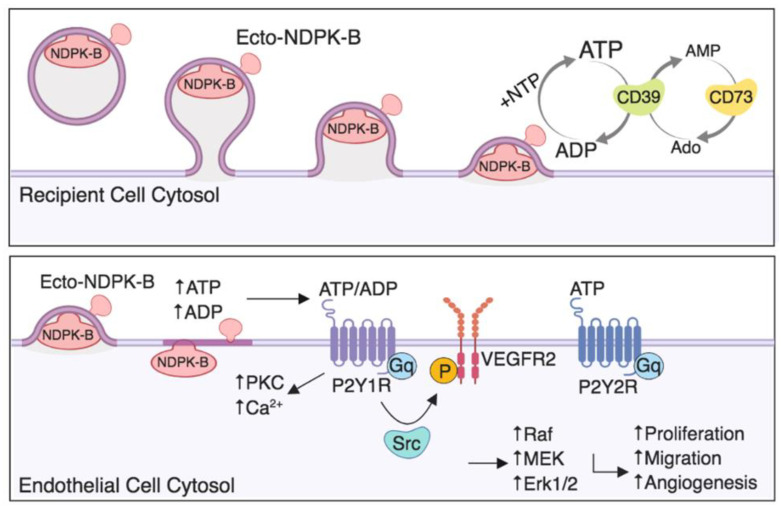
Proposed mechanism of extracellular vesicle-mediated NDPK signaling in endothelial cell activation. Ectopic NDPK-B maintains extracellular ATP and ADP pools by catalyzing phospho-transfer between NTP and NDP substrates. Extracellular ATP and ADP activate the P2Y1 purinergic receptor expressed on vascular endothelium to stimulate known signaling pathways involved in endothelial cell activation, remodeling, and angiogenesis.

## Data Availability

The data presented in this study are available on request from the corresponding author.

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
