# Peer review of "Extracellular Vesicle-Mediated Purinergic Signaling Contributes to Host Microenvironment Plasticity and Metastasis in Triple Negative Breast Cancer"

_ijms, 2021, doi:10.3390/ijms22020597_

Round 1
Reviewer 1 Report
The elucidation of the role of EVs in the metastasis is an important question. The role of NDPKs in vesicles is highly worthy of investigation especially because the mechanism by which NDPK/NM23 contributes to metastasis has not been definitively identified.
Most of the experimental approaches and conclusions in the study are sound. The deficiency is that the authors did not include a genetic approach in the study. The content of the EVs includes, as illustrated in this study, many proteins that can regulate the processes that the authors attribute to NDPK. siRNA or shRNA mediated depletion of NDBK should be included to confirm that the EV activity is due to NDPKs. The siRNA approach has been used in MDA-MB-231 cells to reduce NDPK and the report (Gross et al 2017) is cited here.
Figure 2E Are the multiple lanes in the western replicates? please label
Figure 5 The ZO staining is too weak in the positive control to be able to discern a reduction in experimental samples.
Figure 7 The quality of the graphics is insufficient and the data and labels are unreadable as presented.
The supplemental figures 5 and 6 should contain quantitation.
Author Response
Point 1: The elucidation of the role of EVs in the metastasis is an important question. The role of NDPKs in vesicles is highly worthy of investigation especially because the mechanism by which NDPK/NM23 contributes to metastasis has not been definitively identified.
Response 1: We thank Reviewer 1 for their thorough evaluation of our manuscript and have made efforts to address their salient comments, as described below.
Point 2: Most of the experimental approaches and conclusions in the study are sound. The deficiency is that the authors did not include a genetic approach in the study. The content of the EVs includes, as illustrated in this study, many proteins that can regulate the processes that the authors attribute to NDPK. siRNA or shRNA mediated depletion of NDBK should be included to confirm that the EV activity is due to NDPKs. The siRNA approach has been used in MDA-MB-231 cells to reduce NDPK and the report (Gross et al 2017) is cited here.
Response 2: We strongly agree with the reviewer’s comment that inclusion of a genetic approach would complement the pharmacological studies shown here and robustly define a mechanism for NDPK-mediated signalling in metastasis. Delivery of siRNA or shRNA to deplete NDPK in breast cancer-secreted extracellular vesicles provides an excellent and targeted approach to determine whether the pro-angiogenic activity of EVs is indeed largely attributed to NDPK protein. Here, we observe that pharmacological inhibition of NDPK phosphotransferase activity and blockade of the downstream P2Y1 purinoreceptor independently blunt pro-angiogenic signalling by MDA-MB-231 EVs. Our data also suggest that total phosphotransferase activity in breast cancer-secreted EVs is generated by NDPK, and not by adenylate kinase or other ATP-generating enzymes. Depletion of NDPK protein levels in EVs would lend further vigour to these observations. We have revised the abstract and main text to emphasize this point in the interpretation of results (see revised manuscript, lines 547-557). Unfortunately, we regret to communicate that the suggested experiments cannot be performed at this time as all authors except the corresponding author have since graduated from the institution.
Point 3: Figure 2E Are the multiple lanes in the western replicates? please label
Response 3: The multiple lanes in Figure 2E represent EV lysates generated from three independent isolations, i.e. biological replicates. The legend for Figure 2 has been revised to provide clarification (see revised manuscript, lines 152-153).
Point 4: Figure 5 The ZO staining is too weak in the positive control to be able to discern a reduction in experimental samples.
Response 4: The rationale for performing ZO-1 staining was laid out by the work of Zhou et al. (2014), which showed that MDA-MB-231 exosomes promote cancer cell migration by targeting endothelial tight junctions and reducing ZO-1 expression. However, we acknowledge the reviewer’s comment regarding the weak staining observed in the positive control and have revised Figure 5 to exclude the ZO-1 image panel.
Point 5: Figure 7 The quality of the graphics is insufficient and the data and labels are unreadable as presented.
Response 5: At the reviewer’s request, the quality of the graphics for Figure 7 has been improved and included in the revised text.
Point 6: The supplemental figures 5 and 6 should contain quantitation.
Response 6: At the reviewer’s request, quantitation of the blots shown in Supplemental Figure 6 has been added to the revised Supplemental Figures section. Quantitation of the bioluminescence images shown in Supplemental Figure 5 were presented in Figure 6E as part of the main text.

Reviewer 2 Report
In the manuscript of Duan et al., the authors investigated the role of EV-associated NDPK in modulating the host microenvironment in favor of pre-metastatic niche formation. This manuscript provides very interesting results about the role of EV-associated NDPK-B in lung pre-metastatic niche formation and metastatic outgrowth. However, minor revisions are required in the manuscript.
-Figure 2, flow cytometry data. Gating strategies and the set population % within the gates should be presented on the figures. Furthermore, the panels in 2B (isotype and NDPK-A) are not clear.
-Figure 6: can authors quantify the fluorescence signal of vascular leakage presented in figure 6a? present it in a bar graph format along with the ihc images.
-Figure 7 needs revision for the reader to appreciate the proteomics analysis:
- The text is too small for most labels
- The legend of the heatmap is unclear, what is row min and row max? (Z-scores)
- Please add a legend for the PCA plot or make the groups identifiable otherwise
-Supplementary figure1: legend not clear. Figure part D needs to be clearly specified. What CLSM stands for is not mentioned in the main text nor supplementary.
Author Response
Point 1: In the manuscript of Duan et al., the authors investigated the role of EV-associated NDPK in modulating the host microenvironment in favor of pre-metastatic niche formation. This manuscript provides very interesting results about the role of EV-associated NDPK-B in lung pre-metastatic niche formation and metastatic outgrowth. However, minor revisions are required in the manuscript.
Response 1: We thank Reviewer 2 for their thorough evaluation of our manuscript and have made efforts to address their request for minor revisions.
Point 2: Figure 2, flow cytometry data. Gating strategies and the set population % within the gates should be presented on the figures. Furthermore, the panels in 2B (isotype and NDPK-A) are not clear.
Response 2: As requested, the set population % values and gating strategies described in the Methods Section have been included in the Figure 2 flow cytometry plots. The panels in Figure 2B comparing isotype and NDPK-A labelling suggest a low or near-absent abundance of NDPK-A protein in MDA-MB-231 EVs. Whereas NDPK-A labelling is undifferentiable from the isotype control, NDPK-B labelling is significantly enriched in EVs. Therefore, these observations indicate that the phosphotransferase activities described later in the manuscript are predominantly due to NDPK-B. These observations are further supported by western blot analyses that show significantly higher NDPK-B protein compared to NDPK-A. The main text has been revised to clarify this point to the reader.
Point 3: Figure 6: can authors quantify the fluorescence signal of vascular leakage presented in figure 6a? present it in a bar graph format along with the ihc images.
Response 3: At the reviewer’s suggestion, quantitation of total intensity density of each image has been included in the revised Figure 6.
Point 4: Figure 7 needs revision for the reader to appreciate the proteomics analysis:
- The text is too small for most labels
- The legend of the heatmap is unclear, what is row min and row max? (Z-scores)
- Please add a legend for the PCA plot or make the groups identifiable otherwise
Response 4: Figure 7 has been revised to improve the quality of the images presented. Larger text size has been included when appropriate. In addition, the figure legend has been revised to indicate that the Heatmap row min and max reflect relative log2 fold-change values. The figure legend for the PCA plot includes a description of the group names used in the plot, i.e. Ctrl1-3 represent three PBS-treated mice and 231r1-6 represent six MDA-MB-231 EV treated mice.
Point 5: Supplementary figure1: legend not clear. Figure part D needs to be clearly specified. What CLSM stands for is not mentioned in the main text nor supplementary.
Response 5: We acknowledge the mistake in the Supplemental Figure 1 legend, and have corrected the mislabelling of parts C and D. The description for part C has been revised to describe the appropriate flow cytometry histogram plots shown, while the description for part D has been removed.

Round 2
Reviewer 1 Report
The authors have addressed all the issues and the revised version is acceptible for publication.